# Preliminary Analysis of the Factor Structure, Reliability and Validity of an Obsessive-Compulsive Disorder Screening Tool for Use with Adults in Malaysia

**DOI:** 10.3390/ijerph16234763

**Published:** 2019-11-28

**Authors:** Normah Che Din, Liana Mohd Nawi, Shazli Ezzat Ghazali, Mahadir Ahmad, Norhayati Ibrahim, Zaini Said, Noh Amit, Ponnusamy Subramaniam

**Affiliations:** Health Psychology Program, Faculty of Health Sciences, Universiti Kebangsaan Malaysia, Kuala Lumpur 50300, Malaysia; lianamnawi@gmail.com (L.M.N.); shazli_ezzat@ukm.edu.my (S.E.G.); mahadir@ukm.edu.my (M.A.); yatieibra@ukm.edu.my (N.I.); zainisaid@ukm.edu.my (Z.S.); nohamit@ukm.edu.my (N.A.); ponnusaami@ukm.edu.my (P.S.)

**Keywords:** obsessive-compulsive disorder, reliability, validity, factor analysis, Yale Brown Obsessive-compulsive Scale

## Abstract

This is a preliminary study to examine the factor structure, reliability, and validity of an obsessive-compulsive disorder (OCD) screening tool for use in the Malaysian setting. A total of 199 Malaysian adults were recruited for this study. After cleaning and normalizing the data, 190 samples were left to be analyzed. Principle component analysis using varimax rotation was then performed to examine various factors derived from psychometric tools commonly used to assess OCD patients. The screening tool exhibited three factors that fit the description of obsessions and compulsions from the Diagnostic and Statistical Manual of Mental Disorders—5th Edition (DSM 5), as well as other common symptoms that co-morbid with OCD. The labels given to the three factors were: Severity of Compulsions, Severity of Obsessions, and Symptoms of Depression and Anxiety. Reliability analysis showed high reliability with a Cronbach’s alpha of 0.94, whereas convergent validity of the tool with the Yale Brown Obsessive-compulsive Scale—Self Report demonstrated good validity of r = 0.829. The three-factor model explained 68.91% of the total variance. Subsequent studies should focus on OCD factors that are culturally unique in the Malaysian context. Future research may also use online technology, which is cost-efficient and accessible, to further enhance the screening tool.

## 1. Introduction

Obsessive-compulsive disorder (OCD) is characterized by the presence of compulsions and/or obsessions [1]. Studies have shown that OCD is capable of causing distress, which leads to profound disabilities in occupational functioning as well as social and familial relationships [2]. Individuals with OCD are unable to control their anxious thoughts, leading to their need to engage in ritual behaviors, which have a tremendous impact on their quality of life [3]. Biologically, OCD is said to be caused by an imbalance in the brain of serotonin, which is involved in controlling moods and believed to regulate repetitive behaviors.

Screening for mental health problems, such as OCD, is understood to be one of the most important steps in increasing recognition of the problem, as well as to provide better treatment outcomes [4]. Screening tools have been widely used in clinical settings, including primary care, but are not often used in general population settings. OCD affects 1% to 2% of the population in Malaysia across all ethnic groups and affects both males and females. However, there is still limited published data in Malaysia with regard to the prevalence of OCD, though those which are available mostly cover comorbidity [5]. To date, the screening for mental health problems in Malaysia has used Western criteria based on the Diagnostic and Statistical Manual of Mental Disorders 5th Edition (DSM-5) [1] or the International Classification of Disease 10th Edition (ICD-10) [6].

There are currently no documented mental health screening assessment tools that have been generated entirely and normed in Malaysia, apart from the Saringan Status Kesihatan Mental 20 [7]. As such, this leads to poor assessment of the local factors for mental health disorders within the Malaysian context. The factors leading to poor assessments include: Lack of content, construct, and criterion validity, as well as semantic inaccuracies in the translations into colloquial languages [8]. Apart from the factors above, cultural differences in terms of personal stereotypical beliefs of threats when responding to questions about mental health should also be considered, as these differences have been found to decrease the validity of a measure [9].

The test development followed the best-known standards, such as the “Standards for Educational and Psychological Testing” [10,11]. The scale development process combined steps from different sources [12,13,14,15,16,17,18,19,20]. For example, Trochim [12] and Dimitrov [13] described five steps: (1) Define the measured trait, assuming it is unidimensional; (2) generate a pool of potential Likert items, (preferably 80–100) rated on a five- or seven-point disagree-agree response scale; (3) have the items rated by a panel of experts on a 1–5 scale on how favorable the items measure the construct (from 1 = strongly unfavorable to 5 = strongly favorable); (4) select the items to retain for the final scale; and (5) administer the scale and to some of the responses of all items (raw score of the scale), reversing items that measure something in the opposite direction of the rest of the scale. Furr [14] also described it as a process completed in five steps: (a) Define the construct measured and the context, (b) choose response format, (c) assemble the initial item pool, (d) select and revise items, and (e) evaluate the psychometric properties (see relevant section). Steps (d) and (e) are an iterative process of refinement of the initial pool until the properties of the scale are adequate. Cohen, Swerdlik, and Phillips [15] also described five stages of test construction: (a) Test conceptualization, (b) test construction, (c) test try-out, (d) item analysis, and (e) test revision.

This study initiated the development of a mental health screening tool for OCD symptoms in the Malaysian population to enhance the early detection of OCD symptoms. The development of this screening tool would significantly improve mental health services in Malaysia.

## 2. Materials and Methods

### 2.1. Participants and Procedure

This research includes a cross-sectional study involving convenience and snowball sampling from friends and acquaintances, as well as individuals attending local mental health events. A convenience sampling approach was used to recruit some participants with the aim of applying the instrument across multiple settings and diverse backgrounds [21]. Snowball sampling relies on appointments from initial participants to produce additional participants. This technique is commonly used [22] and is considered economical, efficient, and effective [23].

Sociodemographic data and background were obtained from the participants upon obtaining their consent to participate in this study. The current study adopted an indigenization approach to examine the OCD symptoms experienced by the Malaysian population by combining items from several psychometric tools commonly used in Malaysia, including the Obsessive-Compulsive Inventory-Revised (OCI-R) [24] and Yale–Brown Obsessive-Compulsive Scale (YBOCS) [25]. The questions are in the Malay language and the participants were required to provide ratings when responding to the questions.

Sample estimation for this study adopted the subject to item ratio approach as recommended by Costello and Osborne [26]. Based on their study, factor analysis studies were based on the subject to item ratio of 5:1. In view of this, this study adopted a minimum ratio of five samples per item. The literature contains a variety of recommendations regarding the appropriate sample size to use for conducting a factor analysis [27]. For the most part, these recommendations were presented as either a suggested minimum sample size or a suggested minimum ratio of sample size to number of variables. It is generally accepted that larger samples are better [28,29], but existing recommendations vary [29]. Gorsuch [30] and Kline [28] recommended a minimum sample size of at least 100, whereas Comrey and Lee said 50 is very poor, 100 is poor, 200 is fair, 300 is good, 500 is very good, and 1000 is excellent. Further calculation of sample size using G-Power for multiple regression (using medium effect size = 0.15, α error probability = 0.05, power = 0.95, estimated predictors = 3) is 119.

The inclusion criteria for this study was that all participants were Malaysian citizens age 18 and above. Furthermore, the participants were required to read in Malay. A total of 199 participants were recruited from January to April 2018. Data was collected from the general population. As the present study was intended to be a preliminary study to examine the potential for further development of a psychometric tool, specific demographic variables were not considered at this phase.

### 2.2. Measures

#### Development of Obsessive-Compulsive Screening Tool-Malaysia (OCST-M)

The development of the OCD online screening tool, named the Obsessive-Compulsive Disorder Screening Tool-Malaysia (OCDST-M), was based on the five stages of test construction: (a) Test conceptualization, (b) test construction, (c) test try-out, (d) item analysis, and (e) test revision [15].

The construction of the OCDST-M was based on item pooling. Items were pooled from the Obsessive-Compulsive Inventory-Revised (OCI-R) and the golden standard of OCD screening, which is the Yale-Brown Obsessive-Compulsive Scale-Self Report (YBOCS-SR). As the tool aims to incorporate the theme of *severity* and *comorbid symptoms*, the Likert-type response choices were used. The Likert-type response choices included: (1) Not at all, (2) a little, (3) moderately, (4) a lot, and (5) extremely.

### 2.3. Statistical Analysis

To establish their psychometric properties, the responses from this study were analyzed using exploratory factor analysis (EFA). EFA was selected to explore the relationships among the variables without having *a priori* fixed number of factors. Data was analyzed using the SPSS IBM version 23. The data was subjected to reliability analysis for internal consistency using the Cronbach’s alpha coefficient. Once proven reliable, factor analysis was conducted to determine the factor structure. Certain assumptions needed to be met when conducting EFA (Table 1), based on Hair et al. [31]. Varimax rotation, a recommended rotation technique to use when researchers start exploring the dataset, was employed for factorability. We selected oblique rotation if there was pre-existing evidence that the factors are correlated.

Convergent validity was determined by correlating the screening tool with the criterion gold-standard assessment for OCD—the YBOCS-SR—using the bivariate correlational test.

The factorability of the 20 items was examined using EFA. Before conducting the analysis, the data was screened and examined for any missing values, outliers, and multivariate assumptions, as well as to establish accuracy. Through the descriptive statistics, normality testing was done for the data via skewedness, kurtosis, and z-score. The tests of normality based on Kolmogorov–Smirnov and Shapiro–Wilk exhibited that the data were statistically significant, as attributed by the large sample of this study. The normal distribution of the data was further assessed using skewness and kurtosis, whereby most of the items were ranged within ± 1.

Based on these findings, z-scores were further examined to determine the normality via distribution of scores. Z-scores for each of the items with values more than −2.96 to +2.96 were further examined to check for data abnormalities. Some samples exhibited scores more than −2.96 to +2.96 due to extreme responses on all the questions. The z-score analysis results showed the need to remove nine respondents from the sample. After clearing the faulty data, 190 sets of data remained to be analyzed further.

## 3. Results

### 3.1. Sociodemographic Characteristics

Initially, there were 199 participants in this study. However, nine sets of data from the sample were discarded due to extreme responses. Therefore, analyses of the data were based on the remaining 190 samples. The samples were made up of 95 male participants (50%) and 95 female participants (50%), as shown in Table 2. In terms of ethnicity, there were 118 Malays (62.1%), 52 Chinese (27.4%), 13 Indians (6.8%), and seven participants with other ethnicities (6.6%), as shown in Table 3.

### 3.2. Factorability of Data

Several criteria for the factorability needed to meet the statistical assumptions for a proper estimation of the factor structure. First, all 20 items were correlated at least 0.3 with at least one other item, which suggested reasonable factorability. As the Kaiser-Meyer-Olkin (KMO) measure of sampling adequacy indicated that the strength of the relationships among variables was high (KMO = 0.905), it was acceptable to proceed with the principal component analysis. The Bartlett’s test of sphericity, which determines whether the correlation matrix is an identity matrix, was significant ($x^{2}\left(190\right)=3718.50,\;p<0.001)$, indicating that it was appropriate to use the factor analytic model on this set of data. The diagonals of the anti-image correlation matrix were all found to be over 0.5, which supported the inclusion of each item in the factor analysis. Finally, the communalities were all above 0.3 (see Table 4), which further confirmed that each item shared some common variance with other items. Given these overall indicators, factor analysis was conducted with all 20 items [32].

### 3.3. Three-Factor Model

The 20 items were analyzed via the principal component extraction method which uses the varimax rotation. Three factors with Eigenvalues of over 1.0 were identified. The initial Eigenvalues showed that the first factor explained 51.56% of the variance, the second factor explained 8.94% of the variance, and the third factor explained 8.42% of the variance. Examination of the scree plot revealed a three-factor model that explained 68.91% of the total variance. There were at least four items loaded in each factor, meeting the criteria of the assumptions of factor analysis. After examining the rotated factor solutions, it was decided that the three-factor solution provided a better explanation of the data (Table 5) (Refer Appendix A).

### 3.4. Reliability Analysis

In order to analyze the internal consistency of the OCDST-M, Cronbach’s alpha was examined. The results showed good internal consistency across all items, as well as for each of items within the subscales. All three components resulting from the factor model also showed good reliabilities. Cronbach’s alpha for different dimensions were 0.92 for Factor 1, 0.94 for Factor 2, and 0.84 for factor 3, all with high reliabilities. The total items in the questionnaire showed a Cronbach’s Alpha of 0.94 with high reliability, which showed that the scale is acceptable (Table 6).

### 3.5. Convergent Validity

The YBOCS-SR [33], the gold standard validated assessment, was used to test the convergent validity of the OCDST-M. This scale was designed to assess the severity of obsessions and compulsions within the individual. The current study showed a significant correlation between Y-BOCS-SR and OCDST-M (r = 0.829, n = 190, *p* < 0.01), which demonstrates good validity.

## 4. Discussion

The primary aim of the present study was to develop a prototype OCD symptom screening inventory that reflects the symptomatology of OCD within the Malaysian population. After removing nine items due to extreme responding, principle component analysis revealed a three-factor structure underlying the questionnaire. Results showed good reliability, which indicates the internal consistency of the items. This screening tool also demonstrated excellent convergent validity with YBOCS-SR. The three factors that resulted from the exploratory factor analysis are comparable to the operational definitions of obsessions and compulsions, as well as symptoms likely to comorbid with OCD.

Items loaded on Factor 1 described the experience of obsessions in terms of thoughts, levels of intrusiveness, worries about real-life problems, and feelings of distress. Based on Y-BOCS and DSM-5, these individual items fit the categorization and symptoms of obsessions. Items loaded on this factor also included types of obsessions, such as fear of losing things. This is categorized as miscellaneous obsessions in the Y-BOCS Symptom Checklist. Given the relevance of the items pertaining to severity of the symptoms and types of obsessions, the label *Severity of Obsessions* is recommended for this factor.

Factor 2 was made up of items that assessed the existence of repetitive behaviors, the need to perform these behaviors, levels of distress, and the need to suppress these behaviors. These items fit the description of symptoms of compulsions based on DSM-5 and Y-BOCS. Items loaded on this factor also included types of compulsive acts such as hoarding, cleaning, and aggressiveness. Tools such as Obsessive Compulsive Index (OCI) and Y-BOCS Symptom Checklist would categorize types of compulsive acts in separate OCD themes for better clarity. Given the relevance of the items pertaining to severity of the symptoms and types of compulsions, the label *Severity of Compulsions* is recommended for this factor.

Items relating to depressive and anxiety symptoms, such as eating habits, sleep difficulties, shortness of breath, and feelings of anxiousness, were loaded on to Factor 3. Studies have shown that individuals with OCD symptoms also experience depressive and/or anxiety symptoms, as they feel distressed by the experience [34,35]. Similarly, results of the present study revealed that individuals experiencing OCD symptoms in the Malaysian population presented both depressive and anxiety symptoms, thus suggesting a dimension for *Symptoms of Depression and Anxiety*.

The results of this study showed good reliability ranging from 0.84 to 0.94, which indicates good internal consistency of the items. This screening tool also demonstrated excellent convergent validity with Y-BOCS (r = 0.829, *p* < 0.01).

The study provides insight on the importance of exploring the psychometric properties, reliability, and validity of the OCDST-M. This is one of the first studies that looked into developing an OCD screening tool in the Malay language and focusing on the adult population sample in Malaysia. Although the study recruited different ethnicity samples, all respondents were adults who are well-versed with the Malay language. The Malay language is the national language, which is taught in primary and secondary education levels. Attention was given to the readability of items during the process of development and validation.

The exploratory factor analysis of this study revealed three factors model: *Severity of Compulsions, Severity of Obsessions*, and *Symptoms of Depression and Anxiety*. This is consistent with definitions of obsessions and compulsions from DSM-5, as well as research that shows comorbidities of OCD with depression and anxiety.

This study is a significant step forward in the development of an OCD screening tool attuned to the adult population in Malaysia. Based on this study, the tool can be further enhanced by assessing its sensitivity in distinguishing between clinical and nonclinical samples.

Despite the encouraging statistical findings that supports the OCDST-M as a potentially useful screening tool for the Malaysian OCD population, various confounding variables have yet to be addressed. For instance, a patient’s gender is said to influence symptom manifestation and suicide risk [34]. Longer duration of untreated OCD symptoms is also frequently associated with more severe symptoms and poorer prognosis [35]. These variables were not accounted for by the OCDST-M.

Furthermore, the items loaded on Factor 1, which loaded more items relating to compulsions, indicate that this tool might be heavily skewed toward compulsions as opposed to obsessions. In addition, the highly idiosyncratic nature of obsessions and compulsions forbids any type of self-reporting measure to include an extensive list of the symptoms. As this scale focused mainly on the terms of severity of obsessions and compulsions in general, it did not cover specific OCD symptom presentations such as harm, scrupulosity, and sexual intrusions. Therefore, this tool only provided limited information, which made it unclear what other specific OCD themes the individual might be facing [36].

One other limitation of this approach is the sample of the study, which did not take into account the specific at-risk populations nor help-seeking individuals in clinical settings. Aside from the general population sample, it may be worthwhile to consider recruiting participants who present with complaints or criteria of OCD and individuals with first- or second-degree relatives of patients with OCD symptoms, as well as participants with a certain degree of obsessive-compulsive personality disorder (OCPD) traits. Furthermore, the lack of comparison between a clinical sample and nonclinical sample makes it unclear as to whether this tool is sensitive enough to distinguish between OCD patients and the nonclinical population.

The study used varimax rotation in factor analysis. For future research, it would be appropriate to use promax rotation if factors are correlated and overlap [37]. Furthermore, the study should continue by recruiting another group of participants via online survey and undergo confirmatory factor analysis, as it is more detail and accurate compared to exploratory factor analysis [38].

Screening for mental health problems is understood to be one of the most important steps in increasing recognition of the problem itself and can also provide better treatment outcomes [5]. The use of screening tools has been widely used in clinical settings, including primary care, compared to general population settings [5]. In clinical settings, clinicians are expected to provide extensive screening and anticipatory guidance when patients come in for a visit, usually within a short time frame. Due to the limited time available, the screening process is performed inconsistently, and important problems are often overlooked [39]. One possible alternative is to make use of the Internet with the development of online screening tools. Furthermore, online screening would reach a much wider population, instead of merely those who visit the clinics. Online screening would allow the individuals to seek appropriate care based on feedback they receive about their symptoms and to obtain recommendations for appropriate services [40].

## 5. Conclusions

OCD is undoubtedly a debilitating mental health condition that leads to significant impairment in a patient’s quality of life. This circumstance can be improved through early detection as well as intervention. However, a psychometric instrument to identify OCD that is constructed and normed based on the Malaysian population has yet to be developed. The current study addressed this situation by examining underlying OCD symptoms experienced within the Malaysian population in order to guide the development of the Obsessive-Compulsive Disorder Screening Tool-Malaysia (OCDST-M). The overall results of this study suggested that the preliminary version of this screening tool requires further careful and planned revisions. Based on the findings of this current study, various recommendations for the future OCDST-M were proposed. Despite these limitations, however, this study is a significant step forward for the advancement of mental health psychometric research in Malaysia. The trend toward indigenization is gaining popularity around the world. In the near future, it is hopeful that Malaysia will possess sufficient knowledge and expertise in order to develop its own mental health assessment and diagnostic system that is truly reflective of its population.

## Figures and Tables

**Table 1 ijerph-16-04763-t001:** Assumptions/criteria of factor analysis based on Hair et al. [31].

No.	Matter	Criteria
1	Kaiser-Meyer-Olkin measure of sampling adequacy (KMO)	Range = 0.5–0.7
2	Bartlett’s test of sphericity	<0.05
3	Anti-image correlation matrix of items	0.50 and above
4	Communalities of the variables	0.50 and above
5	Factor loadings	0.30 and above
6	Factors with eigenvalues greater than 1	>1
7	Percentage of variance explained usually 60% or higher	60% and above

**Table 2 ijerph-16-04763-t002:** Percentage of gender in the study.

Age	Number	Percentage (%)
Male	95	50
Female	95	50
Total	190	100

**Table 3 ijerph-16-04763-t003:** Percentage of ethnicity in the study.

Age	Number	Percentage (%)
Malay	118	62.1
Chinese	52	27.4
Indians	13	6.8
Others	7	6.6

**Table 4 ijerph-16-04763-t004:** Item Communalities.

Item	Extraction
Question 1	0.923
Question 2	0.889
Question 3	0.913
Question 4	0.797
Question 5	0.842
Question 6	0.787
Question 7	0.801
Question 8	0.731
Question 9	0.610
Question 10	0.640
Question 11	0.535
Question 12	0.318
Question 13	0.416
Question 14	0.685
Question 15	0.439
Question 16	0.567
Question 17	0.481
Question 18	0.743
Question 19	0.847
Question 20	0.818

**Table 5 ijerph-16-04763-t005:** Factor loadings based on a principle components analysis with varimax rotation for 20 items from the Obsessive-Compulsive Disorder Screening Tool-Malaysia (OCDST-M) (N = 190).

Items	Factor Loadings
**Factor 1**	
Do you engage in repetitive behaviors (e.g., hand washing, ordering, checking) or mental acts (e.g., praying, counting, repeating words silently)?	0.782
Do you feel driven to perform the repetitive behaviors or mental acts in response to an obsession or according to rules that must be applied rigidly?	0.771
Are the behaviors or mental acts aimed at preventing or reducing distress or preventing some dreaded event or situation?	0.751
Are the behaviors or mental acts excessive or unreasonable?	0.672
Do you worry excessively about dirt, germs, or chemicals?	0.761
Are you constantly worried that something bad will happen because you forgot something important, like locking the door or turning off appliances?	0.613
Do you wash yourself or things around you excessively?	0.762
Do you keep many useless things because you feel that you can’t throw them away?	0.592
Have you experienced changes in sleeping or eating habits?	0.528
Do you have to act or speak aggressively even though you really don’t want to?	0.563
*Percentage of total variance (%)*	51.56
**Factor 2**	
Do you experience recurrent and persistent thoughts, impulses, or images?	0.900
Do the thoughts, impulses, or images come from your own mind?	0.886
Do the thoughts, impulses, or images cause you to feel very anxious or distressed?	0.884
Do the thoughts, impulses, or images seem intrusive and inappropriate?	0.790
Do you try to ignore or suppress the thoughts, impulses, or images, or to neutralize them with some other thought or action?	0.843
Are you always afraid you will lose something of importance?	0.525
*Percentage of total variance (%)*	8.94
**Factor 3**	
Do you experience shortness of breath?	0.387
In most days, do you feel sad or depressed?	0.822
In most days, do you feel disinterested in life?	0.875
In most days, do you feel worthless or guilty?	0.841
*Percentage of total variance (%)*	8.42

**Table 6 ijerph-16-04763-t006:** Reliability of Statistics.

Items/Factor	Cronbach’s Alpha	Number of Items
All Items	0.94	20
1	0.92	10
2	0.94	4
3	0.84	5

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
