# Peer review of "Preliminary Analysis of the Factor Structure, Reliability and Validity of an Obsessive-Compulsive Disorder Screening Tool for Use with Adults in Malaysia"

_ijerph, 2019, doi:10.3390/ijerph16234763_

Round 1

Reviewer 1 Report

The authors present a study that developed a Malaysian OCD screening tool including validation, reliability and internal consistency. The paper will well-written, a few points to consider:

-The background is well written however it is too long and can be shortened. Please consider only highlighting pertinent background information in a brief manner while transitioning to the need and objective of your research.

-can the full survey in the Malay language or a link to this survey be made available for those who may need it? Thank you for providing the full English version

-the developed OCDST-M was a self-administered tool correct? How was the online interface set up? Through a website or through a researcher website? Can you suggest future use of this tool based on your experience (online versus paper)?

-given that the manuscript is written in English can you consider providing the factor items in table 5 in English while providing the Malay language version as supplementary material (or vice versa).

Reviewer 2 Report

The paper presents a new tool for OCD and found acceptable validity and reliability indicators from its application. As strengths, some of the indicators, for example alphas and explained variance, are high, also the inclusion of other variables allowed authors to explore contingent validity. also the future usefulness of the instruments is justified.
there are several limitations as well. Some are methodological (sample, procedure, analysis) and others are due to relevant information being missing.

Major Points

-There is not enough literature justifying the procedure used to develop a new instrument. There is a body of literature dedicated to that and many examples of advocated procedures to take into account when deciding to create or adapt and validate an instrument. Authors should engage with this more explicitly in justifying their choices.

-Sample size should be justified as well as the sampling method

-Different ethnicity samples were included in the current sample. Did authors considered any underlying bias that might occur because of this (such as language)?

-Why a varimax rotation has been employed? This needs to be justified, in particular whether factors are not correlated to each others.

-Authors will need and independent sample to confirm the proposed structure. It would be unfair to ask them to recruit new data. For this reason, I would like to ask them to reformulate abstract and title. It is important that the reader does not have the impression that full psychometric properties are examined.

In sum, it is useful to have papers detailing the development of instruments as they are essential to the advancement of research agendas and domains. Despite the limitations, the effort presented here is a step in that direction.

Round 2

Reviewer 2 Report

I thank the authors for improving the manuscript following my commentaires.

I only have two small suggestion to accept the manuscript, that I believe are of interest:

Sample size: Instead of justifying previous literature to use their current sample size, please calculate it through free softwares such as G*Power (as recommended by the APA association): http://www.psychologie.hhu.de/arbeitsgruppen/allgemeine-psychologie-und-arbeitspsychologie/gpower.html

*That will add and extra value to the manuscript.

2. I am partialy satisfied with the explanation that the authors have provided on the rotation method employed. Please, include extra and current literature on the topic. For example, current literature claims that even if varimax rotation is of interest in the previous steps of an EFA (and as described in the response letter by the authors), how factors correlate and an overlap across each others should be visited, in particular for CFA. I recommend to add this piece of information in the discussion section for future lines of research where a confirmatory factor analysis might be carried out, citing literature as follows:

Constant, H. M. R. M., Moret-Tatay, C., Oliveira, M. D. S., Barros, H. M. T., & Ferigolo, M. (2018). CBI-20: Psychometric Properties for the Coping Behaviours Inventory for Alcohol Abuse in Brazil. Frontiers in Psychiatry9, 585.
